# EMC1-dependent stabilization drives membrane penetration of a partially destabilized non-enveloped virus

Parikshit Bagchi, Takamasa Inoue, Billy Tsai*

Department of Cell and Developmental Biology, University of Michigan Medical School, Ann Arbor, United States

**Abstract** Destabilization of a non-enveloped virus generates a membrane transport-competent viral particle. Here we probe polyomavirus SV40 endoplasmic reticulum (ER)-to-cytosol membrane transport, a decisive infection step where destabilization initiates this non-enveloped virus for membrane penetration. We find that a member of the ER membrane protein complex (EMC) called EMC1 promotes SV40 ER membrane transport and infection. Surprisingly, EMC1 does so by using its predicted transmembrane residue D961 to bind to and stabilize the membrane-embedded partially destabilized SV40, thereby preventing premature viral disassembly. EMC1-dependent stabilization enables SV40 to engage a cytosolic extraction complex that ejects the virus into the cytosol. Thus EMC1 acts as a molecular chaperone, bracing the destabilized SV40 in a transport-competent state. Our findings reveal the novel principle that coordinated destabilization-stabilization drives membrane transport of a non-enveloped virus.

*For correspondence: btsai@umich.edu

**Competing interests:** The authors declare that no competing interests exist.

## Introduction

Viruses have evolved sophisticated strategies to penetrate biological membranes to gain entry into host cells and cause disease (*Cosset and Lavillette, 2011*; *Greber, 2016*; *Luisoni et al., 2015*; *Dormitzer et al., 2004*; *Chandran et al., 2002*). While membrane penetration by enveloped viruses is reasonably well characterized, the mechanism by which non-enveloped viruses breach a host membrane remains largely enigmatic. For instance, insights into how the non-enveloped polyomavirus (PyV) penetrates the endoplasmic reticulum (ER) to reach the cytosol are only slowly emerging. Because PyV family members, including the human BK, JC, and Merkel cell PyV, cause debilitating and fatal human diseases, clarifying the precise molecular basis of this decisive infection step should advance our efforts to develop more effective therapies against PyV-induced human diseases (*DeCaprio and Garcea, 2013*; *Jiang et al., 2009*).

Simian virus 40 (SV40) is the archetype PyV, displaying both genetic and structural similarity to human PyVs. Structurally, SV40 consists of 360 copies of the major coat protein VP1 arranged as 72 pentamers (*Stehle et al., 1996*; *Liddington et al., 1991*), with each pentamer encapsulating an internal hydrophobic protein VP2 or VP3 (*Chen et al., 1998*). The assembled pentamers form a viral particle of approximately 45 nm in diameter, which encases the 5 kilobase DNA genome. SV40's disulfide bonds provide critical structural support for the viral particle, while the VP1 C-terminal arms invade neighboring pentamers to further stabilize the overall viral architecture (*Stehle et al., 1996*; *Liddington et al., 1991*). Additionally, calcium ions bound to SV40 strengthens inter-pentamer interactions (*Stehle et al., 1996*; *Liddington et al., 1991*). To infect cells, SV40 traffics from the cell surface to the endoplasmic reticulum (ER) (*Gilbert and Benjamin, 2004*; *Qian et al., 2009*; *Kartenbeck et al., 1989*) where it penetrates the ER membrane to reach the cytosol (*Inoue and*

*Tsai, 2011*). The virus then transports into the nucleus where transcription and replication of the viral genome lead to lytic infection or cellular transformation (*Nakanishi et al., 1996*).

Although SV40 ER-to-cytosol membrane penetration remains enigmatic, a key concept has nonetheless emerged: destabilization of the viral particle is essential to promote membrane translocation. Specifically, upon reaching the ER, protein disulfide isomerase (PDI) family members including PDI and ERdj5 isomerize and reduce SV40's disulfide bonds (*Schelhaas et al., 2007*; *Inoue et al., 2015*); for the murine PyV and possibly JC PyV, another non-catalytically active PDI family member ERp29 locally unfolds the VP1 C-terminal arm (*Magnuson et al., 2005*; *Walczak and Tsai, 2011*; *Nelson et al., 2012*). These reactions disrupt the viral architecture, exposing its hydrophobic proteins VP2 and VP3 (*Magnuson et al., 2005*). The resulting destabilized hydrophobic virus in turn binds to and integrates into the ER membrane (*Norkin et al., 2002*; *Rainey-Barger et al., 2007*; *Kuksin and Norkin, 2012*; *Daniels et al., 2006*; *Geiger et al., 2011*). When the membrane-embedded virus becomes exposed to the cytosol, it likely experiences further destabilization as it is extracted into the cytosol by the cytosolic Hsp70-SGTA-Hsp105 machinery (*Walczak et al., 2014*; *Ravindran et al., 2015*), which is tethered to ER membrane J-proteins including DNAJ C18, as well as DNAJ B12 and B14 (*Walczak et al., 2014*; *Ravindran et al., 2015*; *Goodwin et al., 2011*; *Bagchi et al., 2015*).

Against this framework, we present an opposing yet complementary concept in this manuscript – that viral stabilization is equally important during SV40 ER membrane penetration. Specifically we find that EMC1, the largest component of a poorly-characterized ER transmembrane protein complex called ER membrane protein complex (EMC), is a C18 binding partner that promotes SV40 ER membrane penetration and infection; it also supports BK PyV infection. Surprisingly, EMC1 does so by acting as a molecular chaperone, using its highly conserved predicted transmembrane D961 residue to bind to and stabilize the partially destabilized membrane-penetrating SV40, which likely exposes positively-charged residues on its surface. This interaction braces the virus in a proper conformation so that it can engage the cytosolic extraction machinery and eject into the cytosol. Our results indicate that opposing host-encoded forces coordinately destabilize and stabilize SV40, complementing each other to achieve successful membrane transport. This principle should be broadly applicable to other non-enveloped viruses, where destabilization initiates their membrane transport (*Tsai, 2007*; *Baer and Dermody, 1997*; *Belnap et al., 2000*; *Wiethoff et al., 2005*). Moreover, as the EMC executes important but undefined roles during flavivirus (including Zika, Dengue, and West Nile virus) infection (*Ma et al., 2015*; *Savidis et al., 2016*; *Marceau et al., 2016*; *Zhang et al., 2016*), our findings should have general implications beyond the PyV field.

## Results

### The ER membrane J-protein C18 binds to the EMC1 transmembrane protein

Previous studies found that three ER membrane J-proteins C18, B12, and B14 execute important yet non-overlapping roles during SV40 ER membrane penetration (*Walczak et al., 2014*; *Ravindran et al., 2015*; *Goodwin et al., 2011*; *Bagchi et al., 2015*). We hypothesize that their unique functions may be attributed to their ability to interact with different binding partners, which are all essential in promoting this membrane penetration event. As little is known regarding C18's binding partners, we sought to identify cellular components that interact with this J-protein using a biochemical approach. We first generated a tetracycline inducible 3XFLAG-tagged C18 expressing stable Flp-In 293T-REx cell line [Flp-In 293T-REx (3XFLAG-C18)]. When induced, the level of 3XFLAG-C18 was expressed at a slightly lower level than endogenous C18 (*Figure 1A*). To identify C18-binding partners, cells were induced, mock infected or infected with SV40, then lysed with a mild detergent (Deoxy Big CHAP/DBC) to maintain protein-protein interactions during immunoprecipitation (IP) using anti-FLAG conjugated agarose beads. To control for non-specific binding to the agarose beads, a cellular extract was generated from uninfected parental Flp-In 293T-REx cells, which does not express 3XFLAG-C18 (Flp-In 293T REx). After elution by 3XFLAG peptide, the eluted material was concentrated and subjected to SDS-PAGE followed by silver staining. Importantly, numerous bands appeared in samples prepared from the C18 expressing but not the parental cells (*Figure 1A*, silver stained gel, compare lane 2 to 1). As no significant differences were observed

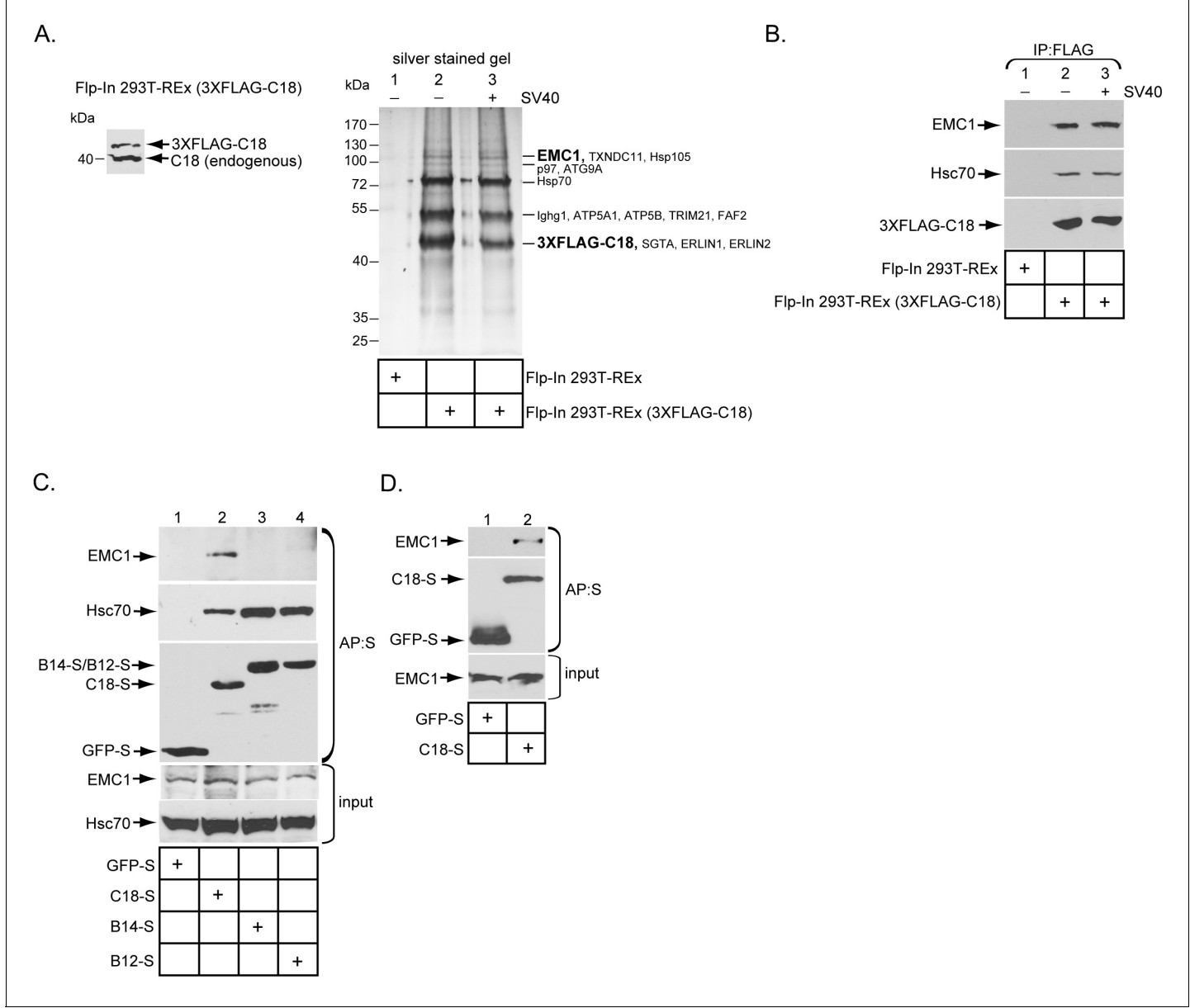

**Figure 1.** The ER membrane J-protein C18 binds to the EMC1 transmembrane protein. (**A**) Expression of 3XFLAG-C18 and endogenous C18 from the Flp-In 293T-REx (3XFLAG-C18) cell line was analyzed by SDS-PAGE followed by immunoblotting with an antibody against C18. 3XFLAG-C18 was immunopurified from induced Flp-In 293T-REx cells infected with SV40 MOI ~50 ('+') or mock infected ('−'). Lysates from the parental cells were used as a negative control. Bound proteins were eluted by 3X FLAG peptide, and subjected to SDS-PAGE followed by silver staining. Bands (indicated on the right) were excised and subjected to mass spectrometry analysis. Protein identities of the bands are listed on the right side of the gel. (**B**) Samples prepared as (**A**) were immunoblotted using the indicated antibodies. (**C**) HEK 293T cells were transfected with the indicated constructs, and after one day of transfection, cells were lysed with 1% Triton X-100 followed by affinity purification with S-agarose beads. The eluted samples were subjected to SDS-PAGE followed by immunoblotting using the indicated antibodies. (**D**) COS-7 cells were transfected with the indicated constructs, and after two days of transfection, cells were lysed with 1% DBC followed by affinity purification with S-agarose beads. The affinity purified samples were subjected to immunoblotting using the indicated antibodies. Please see *Table 1* for the shot –gun mass spectroscopy data.

between the mock and SV40 infected samples (*Figure 1A*, silver stained gel, compare lane 2 to 3), co-precipitated proteins which were reproducibly observed by silver staining were excised from only the SV40 infected sample, and analyzed by mass spectrometry. The results identified potential C18 interacting partners (*Figure 1A*, silver stained gel), including an approximately 110 kDa protein

called ER membrane protein complex subunit 1 (EMC1). EMC1 is predicted to be a single-pass type I ER-resident transmembrane protein that is the largest component of a multiprotein complex called ER membrane protein complex (EMC), which consists of 10 proteins in humans (*Christianson et al., 2012*). Because the EMC was previously implicated in an ER-to-cytosol transport pathway called ER-associated degradation (ERAD) (*Christianson et al., 2012*; *Jonikas et al., 2009*; *Ruggiano et al., 2014*), we focused this study on EMC1 because SV40 similarly undergoes ER-to-cytosol transport. To validate the mass spectrometry data, the same samples were subjected to SDS-PAGE following by immunoblotting using a specific antibody against EMC1, which confirmed the identity of this protein (*Figure 1B*, top panel); as expected, C18 also binds to Hsc70 (*Figure 1B*, middle panel). When the entire precipitated material prepared from C18-expressing and control parental cells were subjected to 'shot-gun' mass spectrometry analyses, in addition to peptides matching EMC1, peptides corresponding to EMC2, EMC3, EMC4, EMC6, and EMC10 were also identified (*Table 1*). In all cases, the number of peptides identified was less than the amount identified for EMC1. This could simply reflect the fact that EMC1 is the largest EMC subunit, or that EMC1 is present in the highest concentration in the precipitated material. Whether Erlin1 and Erlin2, ER membrane factors that appeared to co-precipitate with C18 and also previously implicated in ERAD (*Wang et al., 2009*), control SV40 ER membrane penetration will not be addressed in the current investigation.

To test if EMC1 displays preferential binding to the three ER membrane J-proteins C18, B14, and B12, HEK 293T cells were transfected with either GFP containing a S-tag (GFP-S) or a S-tagged form of the J-proteins. The transfected cells were lysed, followed by affinity purification (AP) with S-agarose beads. When the samples were subjected to SDS-PAGE followed by immunoblotting, only C18 significantly pulled down endogenous EMC1 (*Figure 1C*, top panel, compare lane 2 to 3 and 4). This preferential binding may explain why C18 performs a distinct role from B12 and B14 during SV40 ER membrane penetration (*Bagchi et al., 2015*). The C18-EMC1 interaction was also observed in COS-7 cells (*Figure 1D*), which is a simian cell line derived from the parental CV-1 cells that affords a high transfection efficiency; simian CV-1 cells are typically used to study SV40 entry.

## EMC1 is essential in supporting SV40 and BK PyV infection

To assess if EMC1 promotes SV40 infection, CV-1 cells were transfected with a control siRNA (scrambled) or either of two different EMC1-specific siRNAs (EMC1 siRNA #1 or #2). Immunoblotting of the resulting whole cell extract (WCE) confirmed that EMC1 was largely depleted in cells transfected with the EMC1-specific siRNAs (*Figure 2A*, top panel), while the levels of the single-pass B14/B12/C18 (*Figure 2A*, second, third, and fourth panels) or multi-pass BAP31/Derlin-1 (*Figure 2A*, fifth and sixth panels) membrane proteins were largely unaffected. These are ER membrane factors previously

**Table 1.** Shot-gun mass spectrometry analyses of EMC subunits that bound to C18. 'Shot-gun' mass spectrometry analyses of the FLAG-precipitated material derived from either 3XFLAG-C18 expressing cells or the control parental cells not expressing 3XFLAG-C18 in *Figure 1A*. Only the amounts of unique peptides corresponding to the 10 EMC subunits, as well as C18, are shown.

|       | unique peptide (Flp-In 293T-REx) | unique peptide (Flp-In 293T-REx 3XFLAG-C18) |
|-------|------|------|
| C18   | 0    | 16   |
| EMC1  | 0    | 7    |
| EMC2  | 3    | 4    |
| EMC3  | 0    | 5    |
| EMC4  | 0    | 2    |
| EMC5  | 0    | 0    |
| EMC6  | 0    | 1    |
| EMC7  | 0    | 0    |
| EMC8  | 0    | 0    |
| EMC9  | 0    | 0    |
| EMC10 | 0    | 1    |

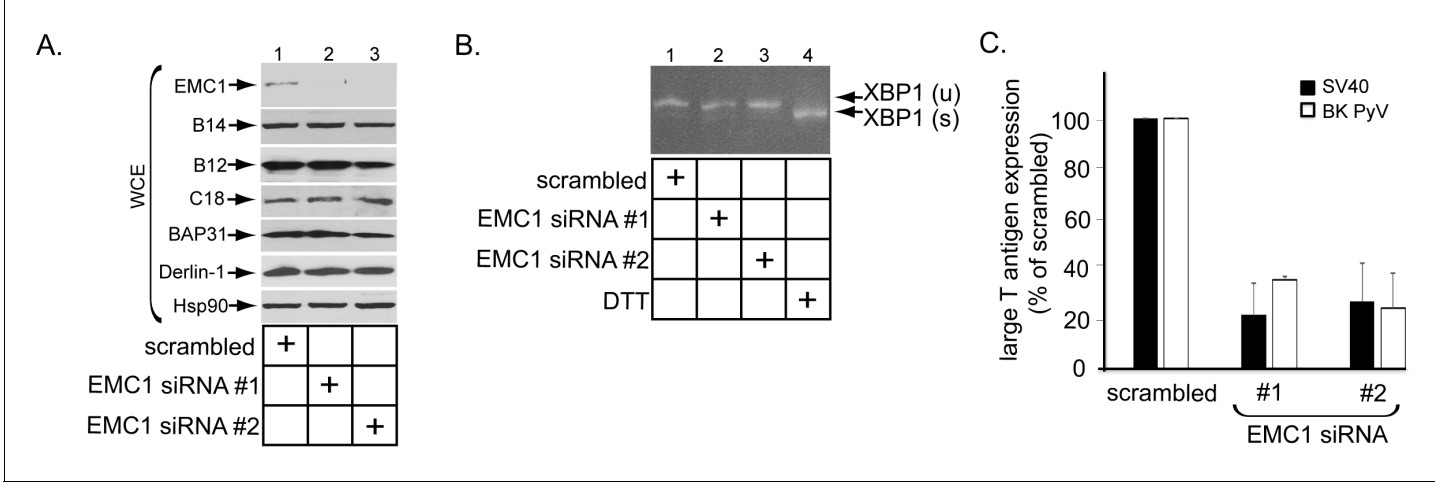

**Figure 2.** EMC1 is essential in supporting SV40 and BK PyV infection. (**A**) CV-1 cells were transfected with either negative control siRNA (scrambled), EMC1 siRNA#1, or #2. Two days after transfection, cells were lysed with 1% Triton X-100 and the resulting whole cell extract (WCE) was subjected to SDS-PAGE followed by immunoblotting using the indicated antibodies. (**B**) CV-1 cells were transfected with either scrambled, EMC1 siRNA#1, or #2 for two days. RNA was isolated from the cells, and RT-PCR was performed to identify splicing of the XBP1 mRNA. Cells treated with DTT served as a positive control. (**C**) CV-1 cells transfected with either scrambled siRNA, or EMC1 siRNA#1 or #2 for 48 hr were infected with either SV40 (MOI ~0.5) for 20 hr (black bars) or BK PyV (MOI ~0.5) for 48 hr (open bars), fixed, and then stained for TAg. Infection was scored using immunofluorescence microscopy. Data are normalized to the scrambled siRNA condition. Values represent the mean ± SD (n ≥ 3). Statistically significant differences (p<0.05) were observed under all EMC1 knockdown conditions when compared to scrambled. Role of EMC subunits 2–10 in SV40 infection is shown in *Figure 2—figure supplement 1*.

The following figure supplement is available for figure 2:

**Figure supplement 1.** Role of EMC subunits 2–9 in SV40 infection.

implicated in SV40 ER membrane penetration (*Schelhaas et al., 2007*; *Geiger et al., 2011*; *Walczak et al., 2014*; *Ravindran et al., 2015*; *Goodwin et al., 2011*; *Bagchi et al., 2015*). We also evaluated whether ER stress was observed when EMC1 was knocked down using the sensitive RT-PCR-based XBP1 splicing assay. We found that depleting EMC1 using either siRNAs did not generate the spliced form of the XBP1 mRNA [XBP1 (s)], in contrast to cells treated with the chemical reductant dithiothreitol (DTT) which triggers ER stress (*Figure 2B*, compare lanes 1–3 to lane 4). These findings suggest that no ER stress induction occurred during EMC1 depletion.

To monitor infection, we scored for presence of the virally-encoded large T antigen expressed in the host nucleus by immunofluorescence imaging as before (*Walczak et al., 2014*; *Ravindran et al., 2015*). Importantly, using this approach we found that depleting EMC1 significantly perturbed SV40 infection (*Figure 2C*, black bars). Moreover, EMC1 knockdown also markedly impaired infection by the human BK PyV (*Figure 2C*, open bars). We conclude that EMC1 is broadly important for promoting PyV infection.

When the other 9 EMC subunits were individually knocked down (*Figure 2—figure supplement 1A*), SV40 infection was attenuated variably (*Figure 2—figure supplement 1B*). These findings raise the possibility that some of the other members of this complex may also have roles during SV40 ER membrane penetration, albeit with different levels of impact.

## EMC1 promotes SV40 ER-to-cytosol membrane penetration

Because EMC1 executes an important role during PyV infection, and that ER-to-cytosol membrane transport is essential during virus infection, we reasoned that the EMC1 is strategically localized at the ER membrane to control this entry step. To test this, we used a cell-based, semi-permeabilized membrane transport assay previously developed in our laboratory designed to monitor ER-to-cytosol membrane transport of SV40 (*Inoue and Tsai, 2011*). In this assay, SV40 infected CV-1 cells are treated with the detergent digitonin that selectively permeabilizes the plasma membrane while

leaving internal membranes (including the ER) intact. The samples were centrifuged to generate a supernatant and pellet fraction. The supernatant (cytosol) should harbor cytosolic proteins and SV40 that has reached the cytosol from the ER, while the pellet (membrane) should contain cellular membranes as well as virus in membranes (including the ER). The integrity of this fractionation procedure was verified by immunoblotting the cytosol and membrane fractions for the presence of cytosolic (Hsp90) or membrane (ER-resident BiP) markers (*Figure 3A*, compare 2nd to 5th panels, and 6th to 3rd panels). Using this method, we found that depleting EMC1 potently decreased the VP1 level (which represents SV40) in the cytosol (*Figure 3A*, top panel; the VP1 band intensity in the cytosol is quantified in *Figure 3B*), indicating that EMC1 is crucial for promoting SV40 ER-to-cytosol transport. We then asked if SV40 arrival to the ER from the cell surface was affected by downregulating EMC1 using a biochemical extraction protocol previously developed in our laboratory(*Inoue and Tsai, 2011*; *Walczak et al., 2014*). In this protocol, the membrane fraction was treated with Triton X-100 to extract detergent soluble virus, which represents ER-localized SV40. This is because only SV40 that successfully reaches the ER can detach from Triton X-100 insoluble lipid rafts and be released into the ER lumen whose content now becomes extractable by this detergent (*Inoue and Tsai, 2011*; *Walczak et al., 2014*). Importantly, using this approach, we found that the level of ER-localized virus was the same between control and knockdown samples (*Figure 3C*, bottom panel; the VP1 band intensity in the ER-localized fraction is quantified in *Figure 3D*), demonstrating that EMC1 does not regulate SV40 trafficking from the plasma membrane to the ER. Hence, the block in SV40 arrival to the cytosol in EMC1-depleted cells reflects a specific role of EMC1 during the viral ER-to-cytosol membrane penetration step. As a negative control, silencing EMC1 did not interfere with ER-to-cytosol transport of another toxic agent cholera toxin's catalytic CTA1 subunit (*Figure 3E*), indicating that EMC1 depletion did not globally affect all ER membrane transport processes.

SV40 is known to reorganize many ER transmembrane proteins such as BAP31 into discrete puncta in the ER called foci, which serve as the viral cytosol entry sites (*Walczak et al., 2014*; *Ravindran et al., 2015*); these transmembrane proteins all sub-serve different functions in guiding the virus across the ER membrane. We now find that SV40 similarly triggers FLAG-tagged wild type (WT) EMC1 (EMC1-FLAG) to mobilize into the foci structure (*Figure 3—figure supplement 1A*). By contrast, the ER membrane protein Sel1L (S/His-Sel1L) does not mobilize into the BAP31-positive foci structure (*Figure 3—figure supplement 1B*), as previously reported (*Bagchi et al., 2015*). Collectively, these results strongly suggest that EMC1 executes a specific function in promoting SV40 ER membrane penetration, consistent with its role in facilitating viral infection.

## The predicted EMC1 transmembrane domain residue D961 plays a critical role during SV40 infection

As a predicted type I transmembrane protein, EMC1's structural organization is expected to be as follows: amino acids 1–21 = signal sequence (cleaved in the mature protein), amino acids 22–958 = luminal domain, amino acids 959–979 = membrane domain, and amino acids 980–993 = cytosolic domain. We hypothesize that EMC1 acts as molecular chaperone, using its transmembrane residues that serve as 'sensors' to recognize and shield exposed polar or hydrophilic residues in partially destabilized SV40 embedded in the ER membrane – this interaction presumably promotes substrate stabilization (see Discussion). In this regard, despite predicted to be within the hydrophobic lipid bilayer, the negatively-charged amino acid D961 and polar amino acid T976 of EMC1 are highly conserved (*Figure 4A*), suggesting an important function. Accordingly, we generated FLAG-tagged D961A and T976A EMC1 mutants to probe their functions during SV40 infection.

We used three different approaches to test whether these mutants behaved similarly to WT EMC1. First, when transfected, these mutants displayed an ER localization pattern as the endogenous ER membrane protein BAP31, similar to transfected WT EMC1 (*Figure 4B*, compare second and third rows to the top row), indicating that D961A and T976A EMC1 are localized to the ER. Second, we asked if the mutants are properly oriented in the ER membrane by performing a deglycosylation assay. EMC1 has three predicted glycosylation sites in its luminal domain. We reasoned that if the EMC1 mutants are inserted in the proper orientation, it should be glycosylated - their ability to be deglycosylated would therefore reflect proper insertion. Accordingly, cell extracts containing WT EMC1-FLAG or the two EMC1 mutants were treated with endoglycosidase H, subjected to SDS-PAGE, followed by immunoblotting using anti- FLAG antibody. We found that both the D961A and T976A EMC1-FLAG mutants displayed a similar faster migration pattern as WT EMC1-FLAG when

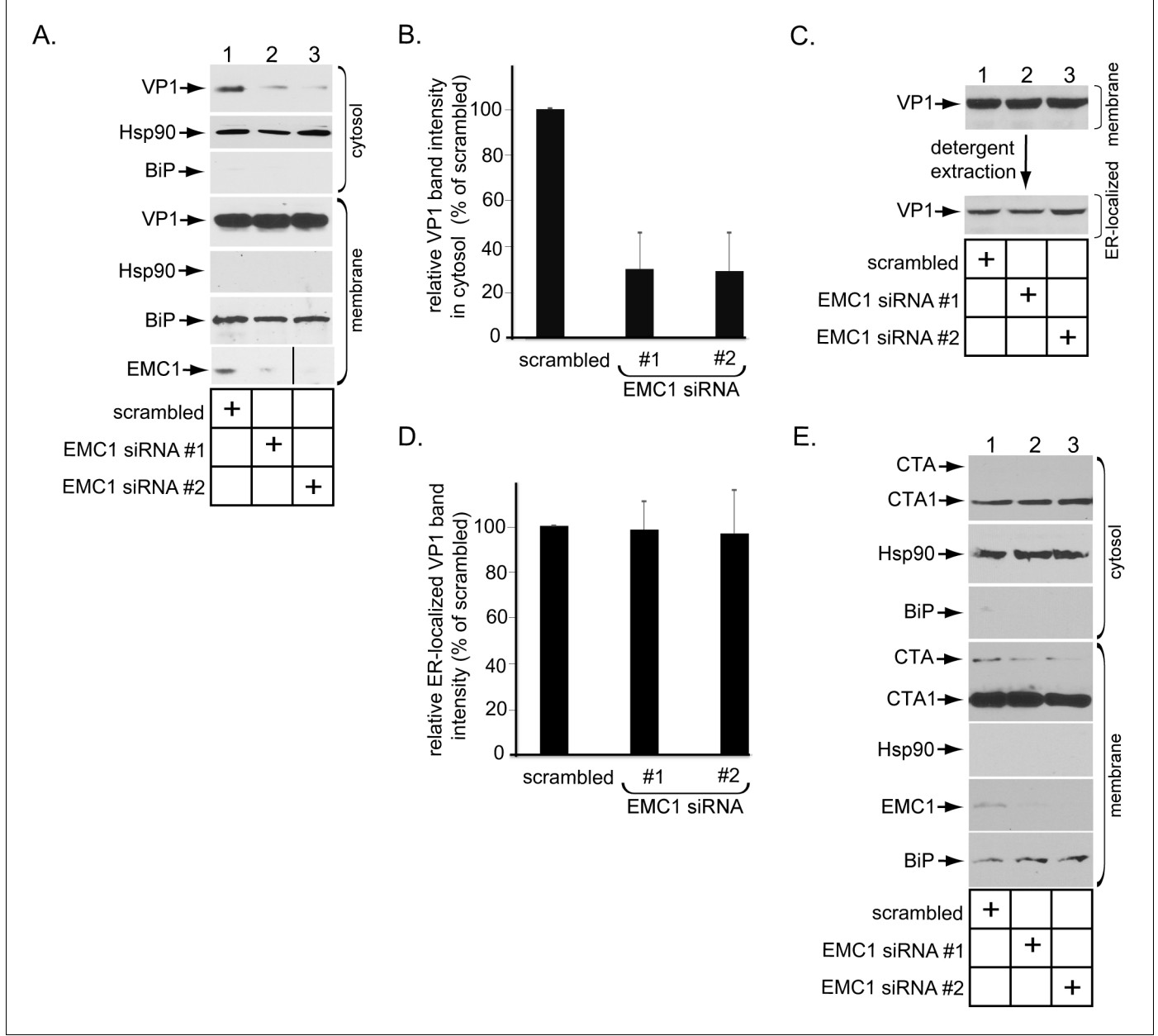

**Figure 3.** EMC1 promotes SV40 ER-to-cytosol membrane penetration. (**A**) CV-1 cells transfected for 24 hr with the indicated siRNAs were infected with SV40 at MOI ~5, harvested 15 hpi, and subjected to the ER-to-cytosol membrane transport assay (see Materials and methods). Cytosolic Hsp90 and ER-resident BiP were used as markers for the cytosol and membrane fractions, respectively. In this assay, the amount of samples loaded in both the cytosol and membrane fractions represent 10% of the total amount in the respective fractions, and they were immunoblotted in parallel with the same exposure time. (The black line indicates that lane 3 of the EMC1 immunoblot was sliced from the same film used for lanes 1 and 2). (**B**) Relative VP1 band intensities from the cytosol fraction from (**A**) were determined using ImageJ (NIH). Data are normalized to scrambled siRNA. Values represent the mean ± SD of three independent experiments, and are statistically significant (p<0.05). (**C**) To isolate ER-localized SV40, the membrane fraction in (**A**) was solubilized in 1% Triton X-100, and the extracted material subjected to SDS-PAGE followed by immunoblotting with the indicated antibodies. (**D**) Relative VP1 band intensities from (**C**) were determined using ImageJ (NIH). Data are normalized to scrambled siRNA. Values represent the mean ± SD of three independent experiments. (**E**) CV-1 cells transfected with the indicated siRNAs were incubated with cholera toxin for 90 min. Cells were harvested, fractionated as in (**A**), and the cytosol and membrane fractions analyzed by using the indicated antibodies. We showed SV40-induced EMC1 foci in *Figure 3—figure supplement 1*.

The following figure supplement is available for figure 3:

**Figure supplement 1.** SV40 induces EMC1 to form foci.

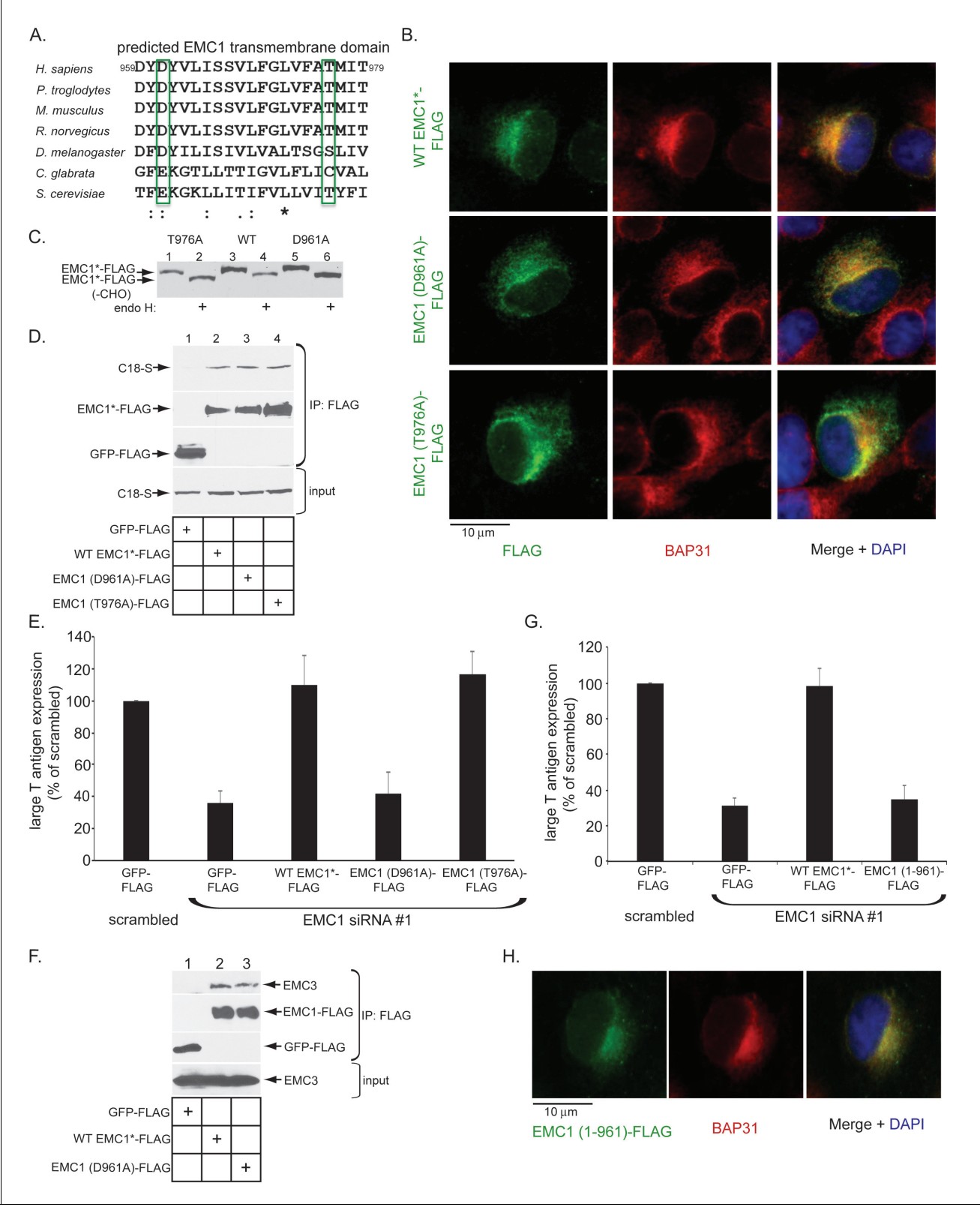

**Figure 4.** The predicted EMC1 transmembrane domain residue D961 plays a critical role during SV40 infection. (**A**) Multiple sequence alignment of the predicted transmembrane domain of EMC1 from human (*H. sapiens*), chimpanzee (*P. troglodytes*), mouse (*M. musculus*), rat (*R. norvegicus*), fly (*D. melanogaster*), haploid yeast (*C. glabrata*), and baker's yeast (*S. cerevisiae*). Highlighted amino acids (in green rectangles) represent highly conserved charged or polar residues. (**B**) CV-1 cells expressing the indicated constructs were stained with an anti-FLAG antibody. BAP31 was used as an ER

*Figure 4 continued on next page*

Figure 4 continued

marker. DAPI positions the nucleus. (C) WT EMC1*-FLAG, EMC1 (D961A)-FLAG, or EMC1 (T976A)-FLAG expressing cells were lysed and treated with endoglycosidase H followed by SDS-PAGE and immunoblotting using a FLAG antibody. (D) Cell lysates generated from HEK 293T cells expressing the indicated constructs were subjected to immunoprecipitation using anti-FLAG antibody coated beads. The bound samples were analyzed by immunoblotting using anti-FLAG and anti-S antibodies. (E) CV-1 cells were transfected with scrambled or the indicated siRNA for 24 hr prior to transfection with the indicated FLAG-tagged constructs for 24 hr. Cells were then infected with SV40 (MOI ~0.5) for 20 hr, fixed, and stained with anti-FLAG and anti-large T antigen antibodies. The percentages of T antigen positive cells were determined only in FLAG-expressing cells by using immunofluorescence microscopy. Values represent means ± SD from three independent experiments. (F) COS-7 cells were transfected with the indicated constructs for 48 hr prior to cell lysis using a buffer containing 1% DBC followed by immunoprecipitation with anti-FLAG antibody. The bound materials were analyzed by immunoblotting using the indicated antibodies. (G) As in E, except cells expressing EMC1 (1-961)-FLAG were also analyzed. (H) As in B, except EMC1 (1-961)-FLAG was used. WT EMC1*-FLAG represents siRNA-resistant EMC1, and all EMC1 mutants were generated using this construct as the template.

treated with this deglycosidase (*Figure 4C*, compare odd to even lanes), suggesting that the orientation of the EMC1 mutants is the same as WT EMC1. Third, we interrogated whether the EMC1 mutants can bind to EMC1's newly identified binding partner C18, and found that they did (*Figure 4D*, top panel, compare lane 2 to lanes 3 and 4). Thus, the D961A and T976A EMC1 mutants are localized to the ER where they are inserted properly and can associate with a binding partner.

To test if the D961 and T976 residues are functionally important during SV40 infection, we performed a knockdown-rescue experiment that our laboratory established previously (*Ravindran et al., 2015*; *Bagchi et al., 2015*). Cells transfected with scrambled or EMC1 siRNA (#1) were subsequently transfected with GFP-FLAG or WT, D961A, or T976A EMC1-FLAG generated in a siRNA #1-resistant EMC1 backbone construct. After SV40 infection, large T antigen expression was scored only in FLAG-expressing cells. Using this protocol, we found that adding back WT EMC1 to cells transfected with EMC1 siRNA #1 fully rescued SV40 infection (*Figure 4E*, compare third to second bar), unambiguously demonstrating that the block in SV40 infection and ER-to-cytosol transport due to EMC1 siRNA #1 treatment is specifically a result of depleting EMC1 and not to off-target effects. Importantly, whereas expressing the T976A mutant also restored infection, expression of the D961A mutant did not (*Figure 4E*, compare fifth to fourth bars). It is unlikely that the D961A mutant failed to form a complex with the EMC because it binds to the EMC3 transmembrane protein, similar to WT EMC1 (*Figure 4F*, top panel, compare lane 3 to 2). These findings thus strongly suggest that the D961 residue of EMC1 plays a specific and crucial role during SV40 infection.

We next asked if a FLAG-tagged EMC1 truncation variant that harbors the luminal domain and including residues up to D961 [EMC1 (1-961)-FLAG] is functional. We found that it was not because expressing this construct in EMC1 knocked down cells did not restore SV40 infection (*Figure 4G*), even though this construct displayed an ER expression pattern (*Figure 4H*). These findings indicate that an intact EMC1 membrane-spanning domain, presumably including residue D961, is required to promote SV40 ER membrane transport.

## EMC1 binds to SV40 during infection and in vitro

Because EMC1 plays important role in SV40 and BK PyV infection (*Figure 2*), as well as in promoting SV40 ER-to-cytosol membrane transport (*Figure 3*), we asked if it binds to SV40. We previously established that SV40 crosses the ER membrane to reach the cytosol approximately 7–8 hpi (*Inoue and Tsai, 2011*). In a time-course experiment, we found that in cells expressing WT EMC1-FLAG, precipitating EMC1 pulled down SV40 only at 7 but not 2 or 15 hpi (*Figure 5A*, top panel, compare lane 2 to 1 and 3). Thus EMC1 binds to SV40 at approximately the time point when the virus is expected to penetrate the ER membrane; this interaction is lost later in the course of infection, perhaps reflecting a completion of SV40 cytosol arrival.

To test if D961 of EMC1 mediates the EMC1-SV40 interaction, we found that the D961A EMC1-FLAG mutant did not efficiently precipitate the virus when compared to WT EMC1-FLAG (*Figure 5B*, top panel, compare lane 2 to 1; the virus binding level is quantified in the bottom bar graph), suggesting that EMC1's D961 residue is critical in supporting virus binding. Additionally, as SV40 co-precipitates poorly with the truncated EMC1 (1-961)-FLAG construct when compared to WT EMC1-FLAG during infection (*Figure 5C*, top panel, compare lane 2 to 1; quantified in the

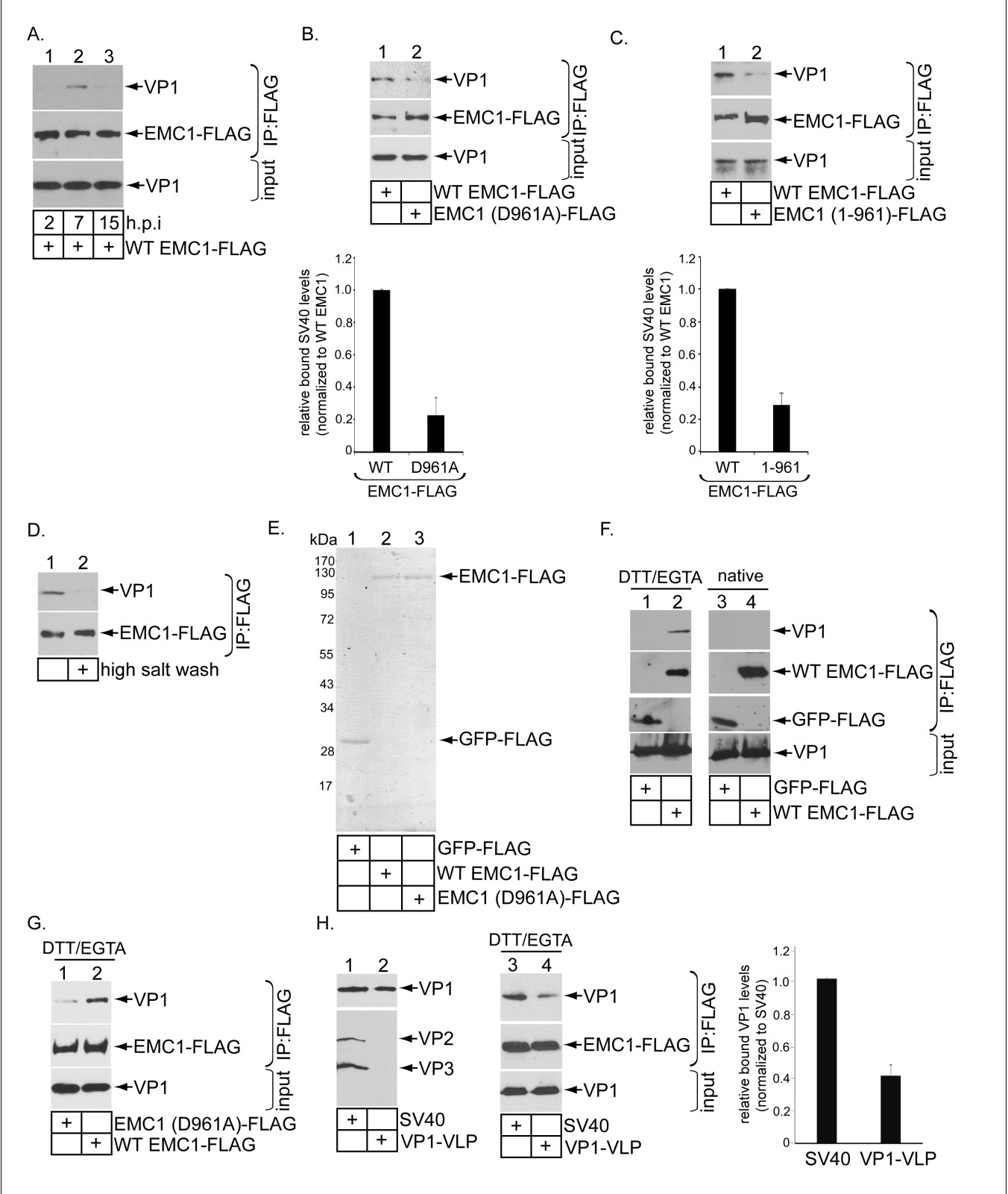

**Figure 5.** EMC1 binds to SV40 during infection and in vitro. (**A**) CV-1 cells expressing WT EMC1-FLAG and subsequently infected with SV40 for the indicated time period were harvested, and the resulting lysate subjected to immunoprecipitation using anti-FLAG antibody coated agarose beads. The bound materials were subjected to SDS-PAGE followed by immunoblotting with the indicated antibodies. (**B**) Cells expressing WT EMC1-FLAG or EMC1 (D961A)-FLAG were infected with SV40 for 7 hr. Cells were processed and analyzed as in (**A**). (right graph) Quantification of the relative levels of

*Figure 5 continued on next page*

*Figure 5 continued*

SV40 bound to WT EMC1 and EMC1 (D961A). Data are normalized to WT EMC1. Values represent means ± SD from three independent experiments. (C) As in B, except EMC1 (1-961)-FLAG was used. (D) The SV40-EMC1 interaction was analyzed at 7 h.p.i. as in (A), except the precipitated material was divided into two sets, with one set subjected to high salt (0.5 M NaCl) wash. The remaining bound material was analyzed by SDS-PAGE and immunoblotting as above. (E) The indicated FLAG-tagged proteins were expressed in and purified from HEK 293T cells, and their purity analyzed by SDS-PAGE followed by Coomassie staining. (F) GFP-FLAG or WT EMC1-FLAG was incubated with either DTT/EGTA treated-SV40 or intact untreated SV40. The FLAG-tagged proteins were precipitated using anti-FLAG antibody coated agarose beads, and the bound materials subjected to SDS-PAGE followed by immunoblotting using the indicated antibodies. (G) WT or EMC1 (D961A)-FLAG was incubated with DTT/EGTA treated-SV40, and the samples analyzed as in (F). (H) (lanes 1–2) VP1-VLP and WT SV40 were analyzed for the presence of VP1, VP2, and VP3. (lanes 3–4) DTT/EGTA-treated SV40 or VP1-VLP was incubated with EMC1-FLAG. The extent of interaction was analyzed as in F. The EMC1-bound VP1 level was quantified in the right graph.

bottom bar graph), the D961 residue must nonetheless act in the context of the rest of EMC1's transmembrane domain to fully support virus interaction. SV40's association with WT EMC1 is sensitive to high salt (0.5 M NaCl) wash (*Figure 5D*, top panel, compare lane 2 to 1), in line with the idea that this binding event is mediated by an electrostatic interaction. These results establish a strong correlation between EMC1's ability to interact with the virus and promote infection, as the D961 residue is crucial to both processes.

EMC1's ability to directly interact with SV40 was next examined using purified components. GFP-FLAG, WT EMC1-FLAG, and D961A EMC1-FLAG were expressed in and purified from HEK 293T cells (*Figure 5E*). Mass spectrometry analyses of the WT and D961A EMC1 samples indicated that no other EMC subunits were present (not shown). This was not surprising because the purifications were intentionally performed in the presence of the harsher detergent Triton X-100, which prevented other EMC subunits from co-purification. Previous studies revealed that SV40 is partially destabilized in the ER (*Inoue et al., 2015*; *Magnuson et al., 2005*; *Walczak and Tsai, 2011*; *Nelson et al., 2012*), an effect that can be mimicked in part by treating purified SV40 with DTT (which reduces the SV40 disulfide bonds) and EGTA (which removes calcium bound to the virus). Indeed, when SV40 was incubated with either GFP-FLAG or WT EMC1-FLAG in the presence or absence of DTT/EGTA, pull down of EMC1 but not GFP precipitated SV40 (*Figure 5F*, top panel, compare lane 2 to 1), demonstrating a direct EMC1-SV40 interaction. By contrast, EMC1 did not interact with untreated native SV40 (*Figure 5F*, top panel, lanes 3 and 4), suggesting that EMC1 recognizes a partially destabilized form of SV40. Moreover, using this in vitro approach, we found that the D961A EMC1 mutant binds to DTT-EGTA destabilized SV40 much less efficiently than WT EMC1 (*Figure 5G*, top panel, compare lane 1 to 2). These data are consistent with the cell-based findings, indicating that D961 of EMC1 plays a critical function in interacting with the virus.

To determine the viral components interacting with EMC1, we used VP1-containing viral-like-particles (VP1-VLP) devoid of VP2 and VP3 (*Figure 5H*, top panel, compare lane 2 to 1). Our results demonstrated that precipitating WT EMC1-FLAG pulled down less VP1-VLP when compared to WT SV40 (*Figure 5H*, top panel, compare lane 4 to 3; quantified in the right graph). These findings indicate that, although EMC1 binds to VP1, the internal proteins VP2 and VP3 also play important roles in supporting this interaction.

## EMC1 stabilizes ER membrane penetrating SV40

What might be the functional significance of the EMC1-SV40 interaction? Because we postulate that EMC1 acts as a transmembrane chaperone, it might use its D961 residue localized within the lipid bilayer to recognize and promote stabilization of membrane-inserted SV40 during the penetration process, as suggested by our finding that the D961 residue plays an important role in EMC1-SV40 binding (*Figure 5*). In this scenario, we envision that depleting EMC1 should destabilize SV40 in the ER, and as a consequence, trigger premature viral disassembly. To test this, we probed ER-localized SV40's structural integrity under the EMC1-depleted condition. We previously established a sucrose sedimentation assay that can distinguish between small/disassembled versus large/intact viral particles (*Inoue and Tsai, 2011*). In this assay, a virus sample is layered on top of a discontinuous sucrose gradient (20–50% sucrose), centrifuged, individual samples collected from top of the gradient, and subjected to SDS-PAGE followed by immunoblotting to detect presence of the virus. Virus

in the bottom fraction corresponds to large/intact particles, while those in higher fractions contain small/disassembled virus. When ER-localized virus derived from control cells was analyzed, the virus was found in the bottom fraction (*Figure 6A*, top panel), suggesting that SV40 remains largely intact despite being partially destabilized in the ER. However, when ER-localized virus derived from EMC1-depleted cells was isolated and subjected to the same procedure, a distinct virus pool is present throughout the higher fractions corresponding to disassembled virus (*Figure 6A*, 2nd panel); these species likely reflect single, as well as oligomeric, VP1 pentamers. Hence a portion of ER-localized SV40 disassembles prematurely in EMC1-depleted cells. As disassembled ER-localized virus is found in the absence of EMC1, this finding suggests that EMC1 acts to stabilize SV40. We note that only a small virus pool was disassembled, consistent with the fact that only a small fraction of the total ER-localized virus attempts to penetrate the ER membrane; most remain in the ER lumen. To rule out the possibility that simply imposing a block in SV40 ER-to-cytosol transport is sufficient to cause viral premature disassembly, ER-localized virus derived from SGTA- or C18-depleted cells, which also display a strong impairment in SV40 ER-to-cytosol transport (*Walczak et al., 2014*; *Goodwin et al., 2011*), was isolated and subjected to the same procedure. However, no disassembled virus appeared in these samples (*Figure 6A*, 3rd and 4th panels). Thus, the appearance of the disassembled virus is not a general consequence of blocking ER membrane transport, but points to a specific role of EMC1 in conferring virus stability.

We used a limited proteolysis approach as an independent strategy to further assess SV40's conformational state when EMC1 is depleted – this is a sensitive method used to detect subtle structural changes that may not be readily revealed by the sucrose gradient sedimentation assay. Using this approach, we found that ER-localized SV40 derived from EMC1-depleted cells is more sensitive to proteolytic digestion by trypsin than virus derived from control cells (*Figure 6B*, compare bottom to top panels). These findings indicate that SV40 is structurally exposed when EMC1 is absent, supporting the sucrose sedimentation data demonstrating that the viral particle experiences premature disassembly without EMC1. These results further support the idea that EMC1 serves to stabilize SV40 during ER membrane penetration.

Whether EMC1 stabilizes the partially destabilized SV40 in vitro was next investigated. When purified SV40 incubated with the control protein bovine serum albumin (BSA) was layered over a discontinuous (20–40%) sucrose gradient, centrifuged, the individual fractions collected, and subjected to SDS-PAGE followed by immunoblotting, the virus was found mostly in the bottom fractions (*Figure 6C*, top panel). When high concentrations of DTT (which reduces the viral disulfide bonds) and EGTA (which removes calcium bound to SV40) were added to completely disrupt inter-pentamer interactions, essentially all the virus now remains in the top fractions, indicating that significant disassembly occurred as expected (*Figure 6C*, second panel). However, if EMC1 was simultaneously incubated in the presence of the destabilizing agents (DTT/EGTA), a significant pool of SV40 sediments to the bottom fraction, representing intact virus (*Figure 6C*, third panel). This finding suggests that EMC1 protects SV40 against disassembly, supporting the notion that it acts to stabilize the virus. However, when SV40 was pre-treated with DTT/EGTA and then incubated with WT EMC1, the virus level that sediments to the bottom fraction decreased significantly compared to when WT EMC1 was simultaneously incubated with DTT/EGTA (*Figure 6C*, compare fourth to third panel). This result demonstrates that WT EMC1 cannot confer protection against viral disassembly once the virus has disassembled. Importantly, the D961A EMC1 mutant did not provide strong protection against DTT/EGTA-induced SV40 disassembly when compared to WT EMC1 (*Figure 6C*, compare fifth to third panel). When quantified, WT EMC1 protected more than 66% of SV40 against DTT/EGTA-triggered disassembly whereas the mutant EMC1 protected approximately 18% of the virus against disassembly, where virus in fractions 1–6 represents disassembled virus and those in fractions 7–9 represents largely intact particles (*Ravindran et al., 2015*) (*Figure 6D*). These results are consistent with the D961A mutant's inability to efficiently bind to SV40 (*Figure 5*) and restore infection in EMC1-depleted cells (*Figure 4*). Hence our cell-based results coupled with the in vitro reconstitution experiments strongly argue that EMC1 imparts a stabilizing force to the virus during the ER membrane penetration process.

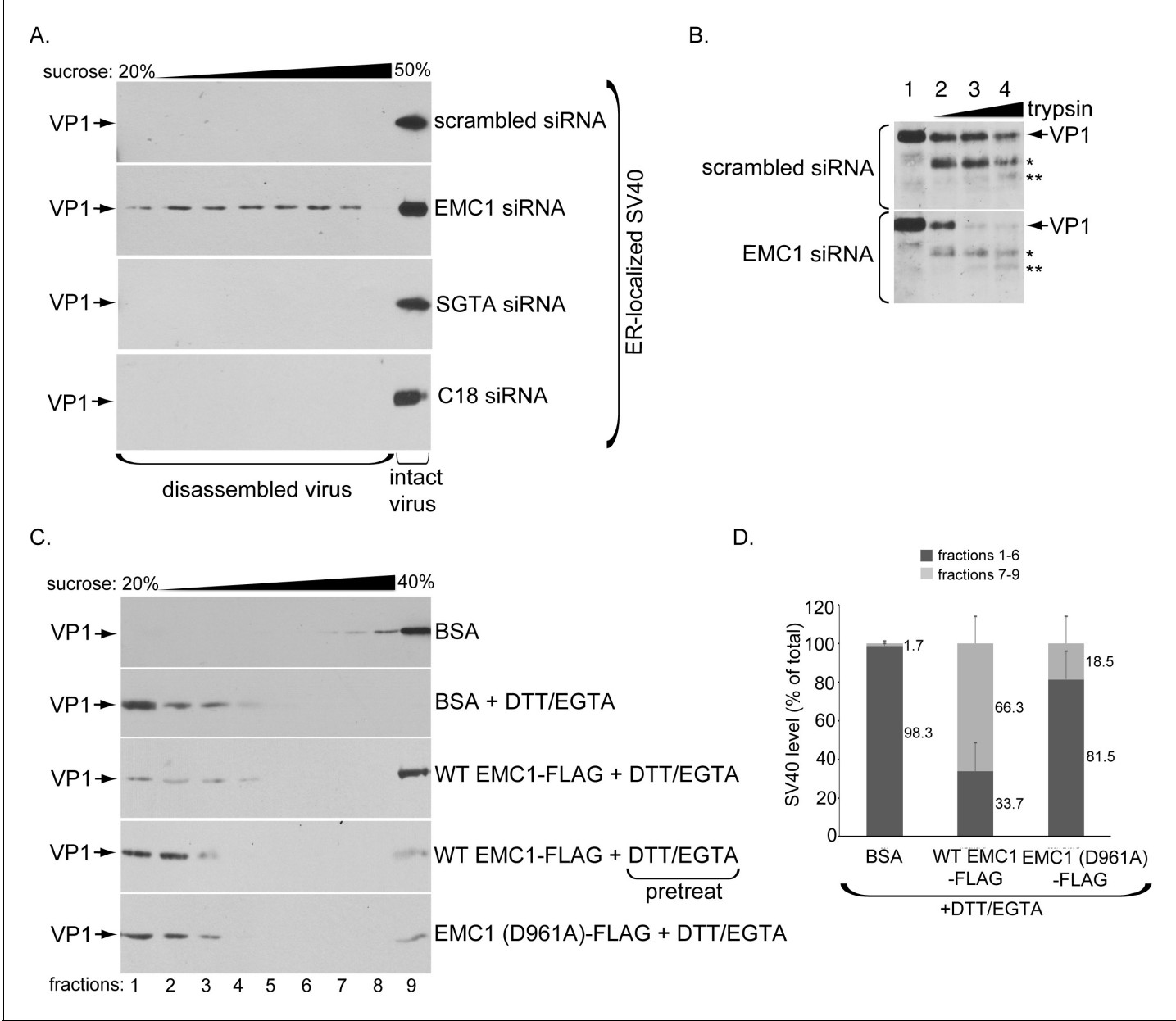

**Figure 6.** EMC1 stabilizes ER membrane-penetrating SV40. (**A**) Extracts containing ER-localized SV40 derived from CV-1 cells treated with the indicated siRNAs were layered on top of a discontinuous sucrose gradient (20–50% sucrose), and centrifuged. Fractions were collected from the top of the gradient and analyzed for presence of SV40 by immunoblotting using VP1 antibodies. (**B**) Extracts containing ER-localized SV40 derived from CV-1 cells treated with the indicated siRNAs were subjected to limited proteolysis using increasing trypsin concentrations. The samples were then subjected to SDS-PAGE followed by immunoblotting using polyclonal VP1 antibodies. * and ** denote degraded VP1 monomers. (**C**) Purified SV40 was incubated with either BSA, BSA + DTT/EGTA, WT EMC1-FLAG + DTT/EGTA, or EMC1 (D961A)-FLAG + DTT/EGTA. Alternatively, SV40 was pre-incubated with DTT/EGTA followed by addition of WT EMC1-FLAG. Samples were layered on top of a discontinuous sucrose gradient (20–40% sucrose) and centrifuged. Fractions were analyzed as in (**A**). (**D**) The amount of VP1 in fractions 1–6 (corresponding to disassembled SV40) and fractions 7–9 (corresponding to largely intact SV40) in samples incubated with BSA, WT EMC1-FLAG, or EMC1 (D961A)-FLAG in the presence of DTT/EGTA in C were quantified.

# EMC1 depletion impairs SV40's engagement with the cytosolic extraction machinery

We reasoned that a prematurely disassembled virus might not assume the proper conformation to properly engage the Hsc70-SGTA-Hsp105 machinery that normally extracts SV40 into the cytosol - this would explain why EMC1 depletion blocks cytosol entry of the virus. To test this possibility, we first asked if Hsc70 is able to capture SV40 when EMC1 is downregulated. Cells expressing S-tagged Hsc70 (Hsc70-S) were transfected with either scrambled or EMC1 #1 siRNA. When Hsc70-S was affinity purified from the resulting WCE, SV40 was only pulled down using extracts from control but not EMC1-depleted cells (*Figure 7A*, top panel, compare lane 2 to 1). By contrast, endogenous SGTA was precipitated under both conditions (*Figure 7A*, second panel, compare lane 1 to 2), suggesting that knocking down EMC1 did not affect the integrity of the Hsc70-SGTA interaction. These data

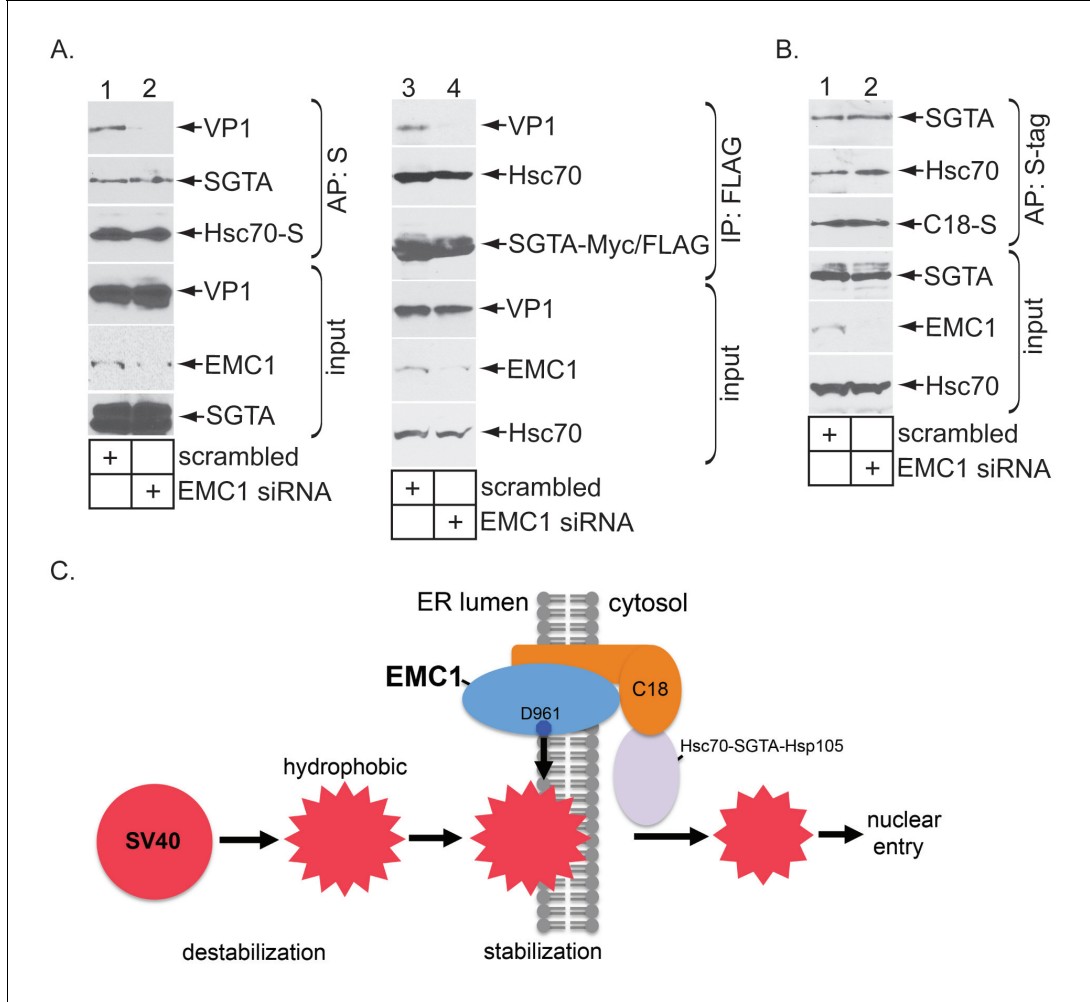

**Figure 7.** EMC1 depletion impairs SV40's engagement with the cytosolic extraction machinery. (**A**) (lanes 1–2) Extracts from SV40-infected CV-1 cells expressing Hsc70-S and treated with the indicated siRNAs were subjected to affinity purification using S-agarose beads. The bound materials were subjected to immunoblotting using the indicated antibodies. (lanes 3–4) As in lanes 1–2, except COS-7 cells were expressing SGTA-Myc/FLAG. (**B**) Extracts from COS-7 cells treated with the indicated siRNAs were subjected to affinity purification using S-agarose beads. The bound materials were subjected to immunoblotting using the indicated antibodies. (**C**) A model depicting how EMC1 stabilizes the partially destabilized SV40 in the ER membrane. When SV40 reaches the ER from the plasma membrane, ER factors destabilize SV40, exposing its internal hydrophobic proteins VP2 and VP3. This generates a partially destabilized hydrophobic viral particle that binds to and integrates into the ER lipid bilayer. Within the bilayer, we propose that EMC1 deploys its D961 residue to engage and stabilize this partially destabilized virus, thereby preventing it from premature disassembly. EMC1-dependent stabilization enables the cytosolic Hsc70-SGTA-Hsp105 machinery to extract the virus into the cytosol in order to complete the membrane transport process.

demonstrate that lack of EMC1, a condition that generates prematurely disassembled SV40 (*Figure 6*), prevents Hsc70 from interacting with the viral particle. A similar result was found for SGTA: EMC1 knockdown prevents (transfected) SGTA from binding to SV40 (*Figure 7A*, top panel, compare lane 4 to 3), but not to endogenous Hsc70 (*Figure 7A*, second panel, compare lane 4 to 3). Together, these findings demonstrate that EMC1 is required to allow SV40 to engage the cytosolic extraction machinery. It is possible that depleting EMC1 precludes the Hsc70-SGTA complex from interacting with C18 – mislocalization of the extraction machinery would effectively prevent it from ejecting SV40 into the cytosol. However, we found that precipitating C18-S pulled down endogenous Hsc70 and SGTA regardless of EMC1's presence (*Figure 7B*, top and second panels, compare lane 2 to 1); silencing EMC1 also preserved the interaction between SGTA and Hsp105 with another J-protein B14 (not shown). Hence, as the extraction machinery remains tethered to ER membrane J-proteins even in EMC1's absence, the simplest explanation for why depleting EMC1 prevents the extraction machinery from capturing SV40 is that the viral particle experienced premature disassembly, thereby precluding its interaction with this machinery. This explanation supports the view that EMC1 braces membrane-embedded SV40 in a membrane penetration-competent state.

## Discussions

Host-encoded forces stabilizing a viral particle are generally exploited to support viral assembly (*Prange et al., 1999*; *Cobbold et al., 2001*). The resulting assembled virus in turn exits the host cell to cause the next round of infection. By contrast, we report here that virus stabilization can also serve a crucial function during viral entry. Specifically, we found that the largest component of the ER membrane complex EMC called EMC1 executes an essential role during SV40 ER-to-cytosol membrane transport, a decisive entry step for this non-enveloped virus.

Our data strongly suggest that EMC1 acts as a molecular chaperone, deploying its highly conserved D961 residue in order to engage the membrane-embedded SV40 and promote the ER membrane penetration event (*Figure 7C*). This residue likely functions within the membrane-spanning domain (or at the membrane border), and must act in concert with the rest of EMC1's transmembrane domain to accomplish its role. Because membrane-penetrating SV40 is partially destabilized, EMC1's interaction stabilizes the overall viral architecture either within the ER lipid bilayer or at the earliest phase of the membrane insertion process. This prevents the viral particle from undergoing premature disassembly. Importantly, stabilization maintains the virus in a transport-competent state, enabling it to be recognized by the cytosolic Hsc70-SGTA-Hsp105 extraction machinery, which completes the membrane transport process by ejecting the virus into the cytosol. Thus, ironically, while host encoded forces that destabilize SV40 are known to promote viral ER-to-cytosol transport (*Ravindran et al., 2015*), an opposing force that stabilizes SV40 appears to play an equally important role during this step. The concept that coordinated interplay between destabilization and stabilization drive PyV entry may have broad implications in illuminating other viral entry mechanisms, including enveloped viruses. For instance, in destabilizing the enveloped virus HIV during entry, host factors also stabilize the viral capsid in a concerted manner in order to prevent its premature uncoating, which otherwise prevents infection (*Guth and Sodroski, 2014*). Thus, there is clearly a precedent for the hypothesis that coordinated destabilization-stabilization is an important (yet poorly-explored) concept during viral entry.

Our finding that EMC1's D961 residue is essential for binding to and stabilizing the partially destabilized SV40 suggests that EMC1 recognizes positively-charged residues exposed in the membrane-inserted virus. Hence, EMC1's transmembrane domain might function as a 'sensor', recognizing exposure of aberrant membrane-spanning residues in substrates. Because VP2 and VP3 are exposed in the membrane-inserted virus (*Magnuson et al., 2005*; *Norkin et al., 2002*; *Rainey-Barger et al., 2007*; *Kuksin and Norkin, 2012*; *Daniels et al., 2006*; *Geiger et al., 2011*), it is tempting to speculate that despite the overall hydrophobic nature of these internal proteins, positively-charged residues of VP2/VP3 may be sensed by EMC1 via D961. This idea is supported by our data that a VP1-VLP lacking VP2 and VP3 displayed less efficient binding to EMC1 when compared to WT SV40 that harbors these internal proteins. The proposed charge-charge interaction would be analogous to sensing of VP2's negatively-charged D17 residue by the positively-charged R transmembrane residues of the ER membrane protein BAP31 (*Geiger et al., 2011*). Positively-charged residues exposed in the coat protein VP1 may also represent the contact point with EMC1, since

VP1-VLP appears to be sufficient in providing some level of interaction with this transmembrane protein.

Precisely how EMC1 stabilizes membrane-embedded PyV is not entirely clear. The postulated exposure of the positively-charged residues in SV40 might cause electrostatic repulsion between the viral particles that would destabilize its structural integrity. Furthermore, displaying positive charges on the viral surface within the hydrophobic ER lipid bilayer creates a highly energetically unfavorable condition. Thus stabilization might reflect a rather passive process in which EMC1 simply shields these charges. Alternatively, EMC1-dependent stabilization may be a more complex active process. For instance, upon recognizing exposure of aberrant residues in membrane-embedded SV40 via EMC1's transmembrane domain, EMC1's luminal domain might subsequently be recruited to provide VP1 inter-pentamer interactions. Our sucrose sedimentation analyses revealed that in infected cells devoid of EMC1, different prematurely disassembled VP1 species, likely corresponding to single and oligomeric VP1 pentamers, are generated (*Figure 6A*). This observation suggests that EMC1 stabilizes inter-pentamer interactions. To enhance VP1 inter-pentamer interactions, EMC1 may exists as oligomers that enable each EMC1 monomer to crosslink to an individual VP1 molecule belonging to different pentamers. Clearly these two possibilities do not have to be mutually exclusive – EMC1 may operate in both manners to shield the presumed exposed charges and to reinforce interactions between the pentamers. In our in vitro analyses, addition of DTT and EGTA also generated VP1 pentamers (*Figure 6B*), whose formation can be largely suppressed by addition of purified EMC1; DTT likely mimics viral disulfide bond reduction that occurs in the ER lumen (*Inoue et al., 2015*), whereas EGTA may imitate loss of Ca2+ from the membrane-embedded virus when it is exposed to the low Ca2+ in the cytosol, as previously hypothesized (*Schelhaas et al., 2007*).

Why might stabilization of membrane-embedded SV40 by EMC1 be important for membrane transport? Our studies revealed that in EMC1-depleted cells, the virus does not engage the cytosolic extraction machinery composed of the Hsc70-SGTA-Hsp105 complex (*Walczak et al., 2014*; *Ravindran et al., 2015*). This finding suggests that the extraction machinery cannot recognize a prematurely disassembled virus in the membrane, but that a largely intact virus is instead required for productive interaction. The specific structural feature of the intact virus recognized by the extraction machinery is unknown, and is an area of investigation worth pursuing. Interestingly, our experiments previously suggested that once the extraction machinery recognizes the membrane-inserted SV40, it has the capacity to disassemble the virus (*Ravindran et al., 2015*) – we postulated that this disassembly reaction might be coupled to the extraction reaction. Thus, while initial recognition of the viral particle by the extraction machinery requires the virus to remain relatively intact, subsequent to recognition, this machinery disassembles it.

In humans, the EMC is composed of 10 subunits (EMC1-EMC10), with EMC1 representing the largest component (*Christianson et al., 2012*). In fact, our data suggest that other EMC subunits, particularly EMC3, EMC4, EMC5, EMC6, EMC7, and EMC10, may also exert roles in supporting virus infection (*Figure 2—figure supplement 1*). Because our findings demonstrate that EMC1 is sufficient to function as a molecular chaperone, whether other EMC subunits can also act in a similar role or execute completely different functions remain to be clarified. The EMC was initially identified in yeast, although its precise function was not fully clarified (*Jonikas et al., 2009*). This complex was proposed to function within an ER quality control process called ERAD, in which misfolded ER proteins are targeted to the cytosol for ubiquitin-dependent proteasomal degradation (*Ruggiano et al., 2014*). This possibility is consistent with the finding demonstrating that different EMC subunits physically associate with other components of the ERAD machinery (*Christianson et al., 2012*). Strikingly, ER-to-cytosol transport of a misfolded substrate during ERAD is highly reminiscent of the pathway SV40 experiences during its ER membrane transport. Another related ER quality control function attributed to the EMC is its ability to support biosynthesis of multi-pass transmembrane proteins, such as the ionotropic acetylcholine receptor (*Richard et al., 2013*) and rhodopsin (*Satoh et al., 2015*). In these cases, the EMC appears to serve as a transmembrane chaperone, promoting the stabilization of the membrane protein folding-intermediates. This idea would agree with our postulate that EMC1 acts as a molecular chaperone to stabilize the membrane-embedded SV40 viral particle. A final but rather distinct function posited for the EMC lies in its ability to physically tether the ER to mitochondria in order to facilitate phospholipid transfer between these two organelles (*Lahiri et al., 2014*). How this function might be linked to EMC's role in ERAD or membrane protein biosynthesis

is difficult to envision, but hints to possibility that the EMC may subserve completely different cellular activities.

Our results illuminating EMC1's function during SV40 membrane transport should have broad significance. For example, a series of recent reports suggest that the EMC plays a critical role during flavivirus (including Zika, Dengue, and West Nile virus) infection (*Ma et al., 2015*; *Savidis et al., 2016*; *Marceau et al., 2016*; *Zhang et al., 2016*). As a wealth of literature has established that replication and assembly of flaviviruses are events in their life cycle associated with the ER membrane (*Ravindran et al., 2016*), it is perhaps not surprising that this ER membrane protein complex might exert a role during their infection. However, despite these reports, the molecular basis by which the EMC controls flavivirus infection remains completely unknown. Whether the EMC functions as a transmembrane chaperone to regulate some aspect of flavivirus replication and assembly requires further investigation. Finally, mutant EMC1 alleles were recently found to be strongly associated with various human neurodevelopmental disorders, including global developmental delay and cerebellar atrophy (*Harel et al., 2016*). Thus, our insights into the molecular function of EMC1 are likely to have impacts beyond the viral pathogenesis field.

## Materials and methods

### Antibodies

Monoclonal SV40 large T antigen antibody (RRID:AB_628305), polyclonal Hsp90 antibody (RRID:AB_2121235) and EMC3 (RRID:AB_10842176) antibody were purchased from Santa Cruz Biotechnology (Santa Cruz, CA). Monoclonal VP1 antibody was kindly provided by Walter Scott (University of Miami). Polyclonal anti S-tag (RRID:AB_444553) and BiP (RRID:AB_732737) antibodies were purchased from Abcam (Cambridge, MA), whereas polyclonal anti-VP1 antibody was a gift from Harumi Kasamatsu (UCLA). Polyclonal DnaJ B14 (RRID:AB_2094414), DnaJ B12 (RRID:AB_2094404), and SGTA (RRID:AB_2188830) antibodies were purchased from Proteintech Group (Chicago, IL). Monoclonal BAP31 (RRID:AB_2537133) and polyclonal Hsc70 (RRID:AB_2544813) antibodies were purchased from Pierce (Rockford, IL), and anti-FLAG tag (RRID:AB_439687) antibody was obtained from Sigma (St Louis, MO). Rabbit anti-EMC1 (RRID:AB_10817224) antibody was purchased from Abgent (San Diego, CA). Polyclonal DnaJ C18 antibody was generated by GenScript (Piscataway, NJ). Polyclonal Derlin-1 antibody was provided by Tom Rapoport (Harvard University). Polyclonal CTA antibody was produced against denatured CTA and generated by EMD Biosciences (San Diego, CA).

### Reagents

CV-1 (RRID:CVCL_0229), COS-7 (RRID:CVCL_0224), and HEK 293T (RRID:CVCL_0063) cells were obtained from ATCC. The Flp-In T-REx 293 cell line (RRID:CVCL_U427) was purchased from Thermo Fisher Scientific (Waltham, MA). Cells were grown in complete DMEM (cDMEM) containing 10% fetal bovine serum, 10 U/ml penicillin, and 10 μg/ml streptomycin (Gibco, Grand Island, NY). Dulbecco's modified Eagle's medium (DMEM), Opti-MEM, 0.25% trypsin-EDTA were purchased from Invitrogen (Carlsbad, CA). Fetal Clone III (FC) was purchased from HyClone (Logan, UT). Complete-mini EDTA-free protease inhibitor cocktail tablets were purchased from Roche. All cell lines used in this manuscript are free of mycoplasma contamination. Anti-FLAG M2 agarose beads and dithiothreitol (DTT) were purchased from Sigma (St Louis, MO). Deoxy Big CHAP (DBC) was purchased from Calbiochem (Billerica, MA), while S-agarose beads was obtained from Novagen (San Diego, CA).

### Preparation of viruses

WT SV40 and VP1-VLP were prepared using OptiPrep gradient system as described previously (*Inoue and Tsai, 2011*). BK virus was kindly provided by Michael Imperiale (University of Michigan).

### siRNA transfection

All Star Negative purchased from Qiagen (Valencia, CA) was used as the control siRNA (labeled as scrambled). Pre-designed siRNAs against EMC1 (ID: 122744 for EMC1 siRNA#1, ID: 122746 for EMC1 siRNA#2) were purchased from Thermo Fisher Scientific (Waltham, MA). Custom siRNA sequences for C18 and SGTA were generated and purchased from Dharmacon (Pittsburgh, PA) or Invitrogen. Their sequences are:

C18 siRNA:       5' GCUAUGAUGAAUACGGAGAUU 3'
                 5' UCUCCGUAUUCAUCAUAGCUU 3'
SGTA siRNA:      5' CCAACCUCAAGAUAGCGGAGCUGAA 3'
                 5' UUCAGCUCCGCUAUCUUGAGGUUGG 3'

Using Lipofectamine RNAiMAX (Invitrogen), 50 nM of EMC1 or C18 siRNA, or 12.5 nM of SGTA siRNA, was reverse transfected into CV-1 or COS-7 cells. Infection or biochemical assays were carried out at 24 or 48 hr post transfection.

## Plasmid constructs

All the plasmids used in this study contain pcDNA3.1 (-) as the vector backbone, and the constructions of these plasmids (B14-S, B12-S, C18-S, GFP-S, Hsc70-S, SGTA-Myc/FLAG and GFP-FLAG) were previously described (*Walczak et al., 2014*; *Ravindran et al., 2015*; *Bagchi et al., 2015*). Protein tags (S- or FLAG-) located at either the N- or C-terminus are depicted as prefix or suffix, respectively. WT EMC1-FLAG was generated from the human EMC1 construct purchased from Harvard Medical School (clone ID HsCD00411858) using the following primers: 5' ATTGTTGCTAGCA TGGCGGCTGAGTGGGCTTCTCGTTTCTGGC 3', 5'ATTGTTGGTACCTTACTTGTCGTCATCGTC TTTGTAGTCTCGCCAGGCCCGATTCAGGAGCTTCACCTGTGCCAGTC 3'. The siRNA-resistant WT EMC1-FLAG (WT EMC1*-FLAG) was generated by introducing silent mutations in the target site of EMC1 siRNA#1. All other EMC1 mutants were generated using WT EMC1*-FLAG as the template. Site directed mutagenesis was performed at D961 or T976 of siRNA resistant WT EMC1-FLAG to generate EMC1 (D961A)-FLAG or EMC1 (T976A)-FLAG. Primers used to generate EMC1 (1-961)-FLAG were 5' GATGACTATAAGGACGACGATGAC 3' and 5' ATAGTCATCCTTCAGAACGTC 3'.

## DNA transfection

50% confluent CV-1 cells were transfected with the indicated plasmid using the FuGENE HD (Promega, Madison, WI) transfection reagent at a ratio of 1:4 (plasmid to transfection reagent; w/v) during the overexpression studies. For COS-7 or HEK 293T cells, polyethylenimine (PEI; Polysciences, Warrington, PA) was used as the transfection reagent. Cells were allowed to express the protein for at least 24 to 48 hr before the experiments were conducted.

## XBP1 splicing assay

Detection of XBP1 splicing was performed as described previously (*Uemura et al., 2009*) using following primers: 5' CGCGGATCCGAATGTGAGGCCAGTGG 3' and 5' GGGGCTTGGTATATATG TGG 3'.

## Immunoprecipitation and affinity purification

Transfected cells were harvested using trypsin, and the cell pellets were washed three times with cold phosphate buffered saline (PBS, Gibco). Washed cells were lysed in HNp buffer (50 mM Hepes pH 7.5, 150 mM NaCl and 1 mM PMSF) with 1% DBC or 1% Triton X-100 at 4°C for 10 min. Cell lysate were clarified by centrifugation at 20,000x g for 10 min at 4°C. The resulting extract was immunoprecipitated with anti-FLAG conjugated agarose beads or affinity purified with S tag protein-conjugated agarose beads for 2 hr at 4°C. Samples were eluted with 1X SDS sample buffer with 1.25% $\beta$-mercaptoethanol (Sigma), and boiled for 10 min at 95°C before subjected to SDS-PAGE and immunoblotting. In the immunoprecipitation experiment described in *Figure 5C*, the sample was subjected to 0.5 M NaCl wash, where indicated.

## Immunopurification and identification of C18 binding partners

Flp-In 293T-REx cells (Invitrogen) transfected with 3XFLAG-C18 were used to immunopurify 3XF-C18. During this procedure, cells selected in media containing blasticidin and hygromycin (Invitrogen) were induced overnight with freshly prepared 0.5 μg/ml tetracycline (Sigma) to express 3XFLAG-C18 at a near endogenous level. 10 μM ganglioside GM1 (Matreya, Pleasant Gap, PA) was supplemented to the culture media prior to infection. Next day, near confluent cells were either mock infected or infected with SV40 (MOI ~50) for 16 hr. 3XFLAG-C18 non-expressing parental Flp-In 293T-REx cells were used as a negative control. Post infection, cells were harvested with cold PBS and centrifuged at 500x g for 5 min. Cells pellets were lysed in 2.5 ml buffer containing 1% DBC in

HNp buffer (50 mM Hepes pH 7.5, 150 mM NaCl and 1 mM PMSF) for 30 min on ice. Lysate was centrifuged at 20,000x g for 15 min, and the resulting supernatant was incubated with anti-FLAG agarose conjugated beads for 2 hr at 4°C. Beads were washed three times with HNp buffer, and the proteins were eluted twice using 3X FLAG peptide (200 µl, 0.25 mg/ml in PBS) (Sigma) for 1 hr at 4°C. The eluted material was concentrated using centrifugal filters (Amicon Ultra 3K, Cork, Ireland), and the concentrated sample was separated on SDS-PAGE and either visualized by silver staining (Invitrogen) or immunoblotted. For mass spectrometry analysis, protein bands were excised from the silver stained gel, and analyzed at Taplin Biological Mass Spectrometry Facility (Harvard Medical School). The data obtained were processed based on the number of unique peptides identified, percentage sequence coverage, and the observed molecular weight.

## ER-to-cytosol membrane transport and ER arrival assays

These assays were performed as in *Inoue and Tsai (2011)*. Briefly, indicated siRNA treated CV-1 cells were infected with SV40 (MOI ~5) for 15 hr. Cells were then lysed in HNp buffer (50 mM Hepes pH 7.5, 150 mM NaCl and 1 mM PMSF) containing 0.1% digitonin at 4°C for 10 min, and separated into supernatant (cytosol) and pellet (membrane) fractions by centrifugation at 20,000x g for 10 min at 4°C. To isolate ER-localized SV40, the pellet fraction was further treated with HNp buffer containing 1% Triton X-100 for 10 min at 4°C, and centrifuged at 20,000x g for 10 min at 4°C. The extracted supernatant material was then dissolved in 1X SDS sample buffer containing 1.25% $\beta$- mercaptoethanol, and boiled for 5 min at 95°C before immunoblotting. To assess cytosol arrival of cholera toxin A1 (CTA1) subunit, CV-1 cells were treated with 10 nM CT (EMD Millipore, Darmstadt, Germany) for 90 min. Cells were harvested and fractionated as above.

## Immunofluorescence microscopy

CV-1 cells were grown in 12 well plate followed by transfection with specific plasmid constructs with FuGene (Promega) for 24 hr. Cells washed with PBS followed by fixation with 1% formaldehyde at room temperature were then permeabilized using 0.2% Triton X-100, and blocked by 5% milk with 0.2% Tween. Primary antibodies were incubated for 1 hr at room temperature, followed by fluorescent conjugated secondary antibodies for 30 min at room temperature. Coverslips were mounted with ProLong Gold (Invitrogen). Images were taken using an inverted epifluorescence microscope (Nikon Eclipse TE2000-E) equipped with 606 and 1006 1.40 NA objective and a Photometrics CoolSnap HQ camera. For knockdown studies, cells were reverse transfected with the desired siRNA using Lipofectamine RNAiMAX (Invitrogen) at the time of cell seeding. ImageJ software (NIH) was used for image processing, analyses, and assembly.

## Knockdown-rescue experiments

CV-1 cells were reverse transfected with specific siRNA using Lipofectamine RNAiMAX (Invitrogen). 24 hr after siRNA transfection, cells were transfected with GFP-FLAG or WT or mutant EMC1 constructs. 24 hr after DNA transfection, cells were infected with SV40 (MOI ~0.5) and at 20 hpi., cells were analyzed by immunofluorescence microscopy using anti SV40 T-antigen and anti-FLAG antibodies. For quantification, T-antigen positive cells were scored in only those cells expressing the FLAG-tagged protein.

## Preparation of purified proteins and partially destabilized virus

For purification of GFP-FLAG, WT and mutant EMC1-FLAG, HEK 293T cells were transfected to express the proteins for 48 hr. Cells were lysed in a buffer containing 1% Triton X-100, 50 mM Tris, 150 mM NaCl, 1 mM EDTA, and protease inhibitors. Cleared lysates were incubated with anti-FLAG agarose beads, and the bound proteins washed extensively with lysis buffer. Proteins were eluted with FLAG peptide overnight and concentrated using centrifugal filters, which also serve to remove any residual FLAG peptides. Optiprep purified SV40 was treated with 5 mM DTT and 10 mM EGTA for 45 min at 37°C to mimic ER-induced conformational changes.

## In vitro binding assays

Binding reactions were carried out in 50 µL PBS containing 100 ng of SV40 (with or without DTT and EGTA) and 200 ng of the purified protein. Reactions were incubated for 30 min at 30°C. Samples

were subjected to immunoprecipitation with anti-FLAG agarose beads, and the precipitated material was eluted with FLAG peptides. The eluted sample was analyzed by SDS-PAGE followed by immunoblotting.

## SV40 disassembly assay

For assessing the viral structural state in infected cells, samples containing ER-localized SV40 derived from specific siRNA-treated CV-1 cells were placed on top of a discontinuous sucrose gradient consisting of 20% to 50% sucrose. Samples were centrifuged at 50,000 rpm in a TLA100 rotor (Beckman Coulter) for 60 min at 4°C, and individual fractions collected from the top of the gradient. Samples were then subjected to immunoblotting using antibodies against VP1. For the in vitro disassembly assay, purified intact SV40 was treated with either BSA (400 ng), BSA + DTT (5 mM)/EGTA(10 mM), WT EMC1-FLAG (200 ng) + DTT/EGTA, or EMC1 (D961A)-FLAG (200 ng) + DTT/EGTA for 30 min at 37°C, and the sample placed on top of a discontinuous sucrose gradient consisting of 20% to 40% sucrose. Alternatively, SV40 was pre-incubated with DTT/EGTA followed by addition of WT EMC1-FLAG before subjected to sucrose gradient centrifugation. Samples were centrifuged at 50,000 rpm in a TLA100 rotor (Beckman Coulter) for 60 min at 4°C, and individual fractions collected from the top of the gradient. Samples were then subjected to immunoblotting using antibodies against VP1.

## Trypsin digestion analysis

ER-localized SV40 was extracted from CV-1 cells infected with SV40 (as described above). The samples were incubated with or without 1, 3, or 10 µg/ml trypsin for 30 min on ice and the reaction was stopped by the addition of 1 mM TLCK for 10 min on ice. The samples were separated by SDS-PAGE followed by immunoblotting with SV40 VP1 polyclonal antibody.

## Statistical analysis

All data obtained from at least three independent experiments were combined for statistical analyses. Results were analyzed using Student's $t$ test. Data are represented as the mean values and error bar represents standard deviation (SD) (n $\geq$ 3) where indicated. $p < 0.05$ was considered to be significant.

## Acknowledgements

We thank Akira Ono (University of Michigan) for critically reading this manuscript. BT is funded by the NIH (AI064296 and GM113722). This study was also partially supported by the Protein Folding Disease Initiative of the University of Michigan Medical School.

## Additional information

### Funding

| Funder | Grant reference number | Author |
| --- | --- | --- |
| National Institute of Allergy and Infectious Diseases | AI064296 | Billy Tsai |
| University of Michigan | Protein Folding Disease Initiative, University Funds | Billy Tsai |
| National Institute of Allergy and Infectious Diseases | GM113722 | Billy Tsai |

The funders had no role in study design, data collection and interpretation, or the decision to submit the work for publication.

### Author contributions

PB, Conceptualization, Data curation, Formal analysis, Investigation, Methodology, Writing—original draft, Writing—review and editing; TI, Conceptualization, Data curation, Formal analysis, Investigation, Writing—review and editing; BT, Conceptualization, Data curation, Formal analysis, Supervision, Funding acquisition, Investigation, Writing—original draft, Writing—review and editing

Author ORCIDs

Billy Tsai, http://orcid.org/0000-0003-2859-1415

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
