## [Decision Letter]

Thank you for submitting your article "EMC1-dependent stabilization drives membrane penetration of a partially destabilized non-enveloped virus" for consideration by *eLife*. Your article has been favorably evaluated by Vivek Malhotra (Senior Editor) and three reviewers, one of whom is a member of our Board of Reviewing Editors. The following individuals involved in review of your submission have agreed to reveal their identity: Elizabeth Miller (Reviewer #2); John Christianson (Reviewer #3).

The reviewers have discussed the reviews with one another and the Reviewing Editor has drafted this decision to help you prepare a revised submission.

Summary:

This study reports the discovery that EMC1, a subunit of the larger EMC (ER membrane complex), is needed for SV40 to efficiently access the cytosol from the endoplasmic reticulum. The authors initially find EMC1 via its interaction with C18, an ER-localized J-domain protein that is thought to act on the cytosolic side at the step of SV40 entry into this compartment. Loss of EMC1 results in decreased SV40 entry into the cytosol, and this activity seems to need D961, an acidic residue near the TM domain of EMC1. The function of EMC1 seems to be the transient stabilization of SV40 after its initial destabilization by reductases in the ER lumen. This stabilization prevents premature disassembly before entering the cytosol and engagement of cytosolic factors. The arguments are built on a series of physical interaction analyses by co-immunoprecipitation (often with tagged proteins over-expressed in cultured cells), loss-of-function studies using siRNA knockdowns, and cytosolic penetration assays.

Essential revisions:

1) One of the central claims in this study is that EMC1 is serving as a transmembrane chaperone. There is relatively little evidence to support this conclusion, except that a key residue in EMC1 (D961) is claimed to be in the TM domain. However, D961 may well be adjacent to the TM domain, rather than really within it. It is important that the authors more rigorously test whether the TM domain of EMC1 is truly necessary for specific engagement of SV40. It would be important to test the lumenal domain (up to at least D961, either attached to a heterologous TMD or in isolation) for activity as it relates to SV40 interaction, stabilization, and membrane penetration. In the absence of a more refined analysis, it is premature to deduce that EMC1 is acting as a membrane chaperone in this process.

2) Thus far, EMC1 has been considered a complex with nine other subunits. Given this foundation, it is important to carefully check that the effects on SV40 are specific to an EMC1-C18 sub-complex rather than the rest of the EMC. It seems plausible that the phenotypes are due to the intact EMC, and that knocking down EMC1 or making the mutant affects some of these interactions or recruitment of virus to EMC, and is not necessarily indicative of EMC1-virus direct interactions. In this scenario, the experiment with purified EMC1 would have contaminating EMC, which is easy to miss because they are much smaller and stain less well. Thus, the authors may wind up being wrong about EMC1 directly interacting with virus (a central claim of this study) or that it works in isolation without the remainder of the complex. This important issue should be examined in more depth. A more thorough accounting of the mass spectrometry data and purified EMC1 preps for the other EMC subunits, along with some control experiments knocking down other EMC subunits, seems essential to validating some of the primary claims in this paper.

3) One concern that arose during discussions among the referees is that the level of mechanistic insight into the role of EMC1 during SV40 membrane penetration was relatively modest and mostly limited to physical interaction by co-IPs. The paper would be substantially strengthened by greater insight into what exactly EMC1 recognizes or binds to on the virus. A number of regions have been characterized as becoming exposed upon SV40 destabilization, and some information about what is being detected by EMC1 would greatly strengthen the evidence for its role in recognition of destabilized virus. At the very least, it would be useful to provide evidence that EMC1 is indeed directly engaging the virus by visualizing co-localization at the site of egress from the ER (for example, as was done with BAP31 by Geiger et al., 2011, NCB).

[Editors' note: further revisions were requested prior to acceptance, as described below.]

Thank you for resubmitting your work entitled "EMC1-dependent stabilization drives membrane penetration of a partially destabilized non-enveloped virus" for further consideration at *eLife*. Your revised article has been favorably evaluated by Vivek Malhotra (Senior Editor) and the Reviewing Editor.

The Reviewing Editor has concluded that you have addressed the concerns raised during the first round of review with only (relatively minor) exception: the experiment designed to show EMC1 co-localized to foci during SV40 infection (Figure 3—figure supplement 1) lacks a control to illustrate that an unrelated ER protein is not enriched in such foci. In other words, it seems important to show that this bright spot is not simply total ER enriched in this area.

---

## [Author Response]

*Essential revisions:*

*1) One of the central claims in this study is that EMC1 is serving as a transmembrane chaperone. There is relatively little evidence to support this conclusion, except that a key residue in EMC1 (D961) is claimed to be in the TM domain. However, D961 may well be adjacent to the TM domain, rather than really within it. It is important that the authors more rigorously test whether the TM domain of EMC1 is truly necessary for specific engagement of SV40. It would be important to test the lumenal domain (up to at least D961, either attached to a heterologous TMD or in isolation) for activity as it relates to SV40 interaction, stabilization, and membrane penetration. In the absence of a more refined analysis, it is premature to deduce that EMC1 is acting as a membrane chaperone in this process.*

As requested, we generated an EMC1 construct spanning residues 1-961, and found that it cannot restore SV40 infection in EMC1-depleted cells (Figure 4) despite displaying an ER localization pattern (Figure 4), nor bind to the virus efficiently (Figure 5). These observations are similar to the D961A EMC1 point mutant, which also fails to restore infection (Figure 4) and binds to the virus less effectively (Figure 5). Thus, D961 of EMC1 functions during SV40 ER membrane penetration either within the membrane-spanning domain or at the membrane border, and must act coordinately with the rest of EMC1’s transmembrane domain to accomplish its role. Because membrane-penetrating SV40 is partially destabilized, EMC1’s interaction stabilizes the overall viral architecture either within the ER lipid bilayer or at the earliest phase of the membrane insertion process. We now state this in the revision.

Despite these new findings, we recognize this reviewer’s skepticism regarding the term “transmembrane chaperone”, since we have not definitively demonstrated that D961 resides in the lipid bilayer. As such, we have changed the term “transmembrane chaperone” to simply “molecular chaperone”.

*2) Thus far, EMC1 has been considered a complex with nine other subunits. Given this foundation, it is important to carefully check that the effects on SV40 are specific to an EMC1-C18 sub-complex rather than the rest of the EMC. It seems plausible that the phenotypes are due to the intact EMC, and that knocking down EMC1 or making the mutant affects some of these interactions or recruitment of virus to EMC, and is not necessarily indicative of EMC1-virus direct interactions. In this scenario, the experiment with purified EMC1 would have contaminating EMC, which is easy to miss because they are much smaller and stain less well. Thus, the authors may wind up being wrong about EMC1 directly interacting with virus (a central claim of this study) or that it works in isolation without the remainder of the complex. This important issue should be examined in more depth. A more thorough accounting of the mass spectrometry data and purified EMC1 preps for the other EMC subunits, along with some control experiments knocking down other EMC subunits, seems essential to validating some of the primary claims in this paper.*

As requested, we have now done the following:

A) We analyzed purified WT and D961A EMC1 by mass spectrometry, and found no other EMC subunits in the preparations. Specifically, whereas we found 46 peptides corresponding to EMC1 using the WT EMC1 prep and 42 peptides corresponding to EMC1 using the D961A EMC1 prep, we did not identify any peptides corresponding to other EMC subunits. These findings are not surprising because we intentionally used the harsh detergent Triton X-100 (1%) during purification, which would preclude co-purification of other EMC subunits. Therefore, any difference in activity between the WT and D961A EMC1 preps in our experiments is due to the specific EMC1 mutation.

B) In addition to analyzing individual band slices, we now analyzed the entire 3XFLAG-C18-precipitated material using “shot-gun” mass spectrometry, using mock precipitated sample derived from cells not expressing 3XFLAG-C18 as a control. It should be noted that, to identify potential C18-binding partners, the gentle detergent deoxyBigChap (1%) was used in order to preserve weak protein-protein interactions. Our new analyses revealed that, in addition to peptides matching EMC1, peptides corresponding to EMC2, EMC3, EMC4, EMC6, and EMC10 were also identified (Table 1). In all cases, the number of peptides identified was less than the amount identified for EMC1. This could simply reflect the fact that EMC1 is the largest EMC subunit, or that EMC1 is present in the highest concentration in the precipitated material.

C) Regardless of the amount of other EMC subunits that co-precipitated with C18, we decided to perform more comprehensive analyses by individually knocking down all of the other 9 EMC subunits (Figure 2—figure supplement 1), and assessing the consequences on viral infection. We now find that depleting EMC4, EMC6, and EMC7 significantly decreased virus infection, depleting EMC3, EMC5, and EMC10 partially impaired infection, while depleting EMC2, EMC8, and EMC9 (postulated cytosolic components of the EMC) had no effect on infection (Figure 2—figure supplement 1). These analyses clearly raise the possibility that other EMC subunits may be involved in SV40 infection, potentially in conducting SV40 across the ER membrane. However, whether other EMC subunits act as molecular chaperones similar to EMC1 or execute completely different functions during SV40 entry remain to be clarified.

*3) One concern that arose during discussions among the referees is that the level of mechanistic insight into the role of EMC1 during SV40 membrane penetration was relatively modest and mostly limited to physical interaction by co-IPs. The paper would be substantially strengthened by greater insight into what exactly EMC1 recognizes or binds to on the virus. A number of regions have been characterized as becoming exposed upon SV40 destabilization, and some information about what is being detected by EMC1 would greatly strengthen the evidence for its role in recognition of destabilized virus. At the very least, it would be useful to provide evidence that EMC1 is indeed directly engaging the virus by visualizing co-localization at the site of egress from the ER (for example, as was done with BAP31 by Geiger et al., 2011, NCB).*

As requested, we have performed 2 additional experiments to address the concerns here:

A) In the original manuscript, we speculated that the internal proteins VP2 and VP3 might be involved in binding to EMC1. We now used VP1-containing viral-like-particles (VP1-VLP) devoid of VP2 and VP3 (Figure 5, top panel, compare lane 2 to 1), and demonstrate that precipitating WT EMC1-FLAG pulled down less VP1-VLP when compared to WT SV40 (Figure 5, top panel, compare lane 4 to 3; quantified in the right graph). These new findings indicate that, although EMC1 binds to VP1, the internal proteins VP2 and VP3 also play important roles in supporting this interaction.

B) SV40 is known to reorganize many ER transmembrane proteins such as BAP31 into discrete puncta in the ER called foci, which serve as the viral cytosol entry sites, as indicated by this reviewer. These transmembrane proteins all sub-serve different functions in guiding the virus across the ER membrane. In the revision, we now demonstrate that SV40 similarly triggers WT EMC1-FLAG to mobilize into the foci structure (Figure 3—figure supplement 1). These results support the idea that EMC1 executes a specific function in promoting SV40 ER membrane penetration.

[Editors' note: further revisions were requested prior to acceptance, as described below.]

The Reviewing Editor has concluded that you have addressed the concerns raised during the first round of review with only (relatively minor) exception: the experiment designed to show EMC1 co-localized to foci during SV40 infection (Figure 3—figure supplement 1) lacks a control to illustrate that an unrelated ER protein is not enriched in such foci. In other words, it seems important to show that this bright spot is not simply total ER enriched in this area.

We have now included the last requested negative control experiment, in which we demonstrate that the ER membrane protein Sel1L does not form SV40-induced foci (Figure 3—figure supplement 1). We note that we had previously published this negative control experiment (Bagchi et al., 2015, Journal of Virology, 89, 4058-68). Moreover, we also previously published that the ER membrane protein Hrd1 (which forms a complex with Sel1L) also does not form virus-induced foci (Walczak et al., 2014, PLoS Pathogen, 10, e1004007).